# New Insights into Geometric Morphometry Applied to Fish Scales for Species Identification

**DOI:** 10.3390/ani14071090

**Published:** 2024-04-03

**Authors:** Francesca Traverso, Stefano Aicardi, Matteo Bozzo, Matteo Zinni, Andrea Amaroli, Loris Galli, Simona Candiani, Stefano Vanin, Sara Ferrando

**Affiliations:** 1Department of Earth, Environmental, and Life Sciences, University of Genoa, Corso Europa, 26, 16132 Genoa, Italystefano.aicardi94@libero.it (S.A.); matteo.bozzo@unige.it (M.B.); matteo.zinni89@gmail.com (M.Z.); andrea.amaroli@unige.it (A.A.); loris.galli@unige.it (L.G.); candiani@unige.it (S.C.); stefano.vanin@unige.it (S.V.); 2National Biodiversity Future Center (NBFC), Piazza Marina, 61, 90133 Palermo, Italy

**Keywords:** landmark, outline, momocs, geomorph

## Abstract

**Simple Summary:**

The identification of fish species from a single dermal scale is intriguing in various contexts, including ecology, commerce/forensics, archaeology, and others. While molecular methods serve as the gold standard for species attribution in some cases, they may not always be feasible and, moreover, they are expensive and need specialized personnel. The prospect of attributing fish species through the analysis of dermal scale shape presents a cost-effective alternative that, after the development of dedicated software, could make the process nearly fully automated. The term “geometric morphometry” encompasses a range of techniques designed to quantify and statistically analyze shapes. In this study, after reviewing previous literature on geometric morphometry applied to fish scales, two distinct methods of geometric morphometry were employed on scales from five different fish species. The advantages of both methods were compared. While one method, referred to as landmark-based and commonly utilized in previous literature, was optimized through the introduction of technical enhancements. The second method, known as outline-based and less prevalent in the literature, demonstrated superior performance and holds promise for future automation.

**Abstract:**

The possibility of quick and cheap recognition of a fish species from a single dermal scale would be interesting in a wide range of contexts. The methods of geometric morphometry appear to be quite promising, although wide studies comparing different approaches are lacking. We aimed to apply two methods of geometric morphometry, landmark-based and outline-based, on a dataset of scales from five different teleost species: *Danio rerio*, *Dicentrarchus labrax*, *Mullus surmuletus*, *Sardina pilchardus*, and *Sparus aurata*. For the landmark-based method the R library “geomorph” was used. Some issues about landmark selection and positioning were addressed and, for the first time on fish scales, an approach with both landmarks and semilandmarks was set up. For the outline-based method the R library “Momocs” was used. Despite the relatively low number of scales analyzed (from 11 to 81 for each species), both methods achieved quite good clustering of all the species. In particular, the landmark-based method used here gave generally higher R^2^ values in testing species clustering than the outline-based method, but it failed to distinguish between a few couples of species; on the other hand, the outline-based method seemed to catch the differences among all the couples except one. Larger datasets have the potential to achieve better results with outline-based geometric morphometry. This latter method, being free from the problem of recognizing and positioning landmarks, is also the most suitable for being automatized in future applications.

## 1. Introduction

In 2020, the combined production of fisheries and aquaculture achieved an unprecedented milestone, totaling 214 million tons and being valued at approximately USD 424 billion [1]. The output of aquatic animals in 2020 surpassed the 1990s average by more than 60%, significantly surpassing the rate of global population growth. The consumption of seafood has reached a historic high, with each person consuming around 20.2 kg in 2020—more than double the rate 50 years ago. On a global scale, seafood contributes to approximately 17% of animal protein, with some countries in Asia and Africa exceeding 50% [1]. Live, fresh, and chilled fish are still the preferred choices for human consumption. However, accidental or deliberate substitutions and mislabeling—especially of species subject to restrictions, either due to overexploitation, surpassed quotas, or endangered status—are common in many countries and they are performed to enhance profitability. Such substitutions are undesirable, carrying potential economic and ecological consequences, and may compromise food safety [2]. Correct species identification is crucial in preventing and investigating food fraud and illegal trade. While molecular analyses are undoubtedly the gold standard for fish species identification, they are not always feasible due to their cost and the need for specialized laboratories.

Among both marine and freshwater fish, the most abundant group in terms of number of species and human consumption is that of teleosts [1,3] such as tuna, carp, herring, anchovy, etc.

The skin of teleost species often presents epithelial appendages called elasmoid scales. Distinguishing teleost species relying only on scales shape can be a useful tool during fish inspection carried out at markets and on ships. Moreover, it can offer the possibility to study the diets of various organisms when scales are found in their stomach [4,5,6] and it can also be helpful for the study of past human diet, analyzing fish scales from archaeological excavations [7]. Geometric morphometry is a non-disruptive, fast, and economic method that is useful for shape discrimination and particularly applicable in fish identification for scientific purposes [8]. Landmark-based geometric morphometry (LM-based GM), applied to fish scales, has proved to be useful to discriminate different genera [9], species [10,11], populations [12,13], and different phenotypes coexisting in the same environment [14], to ensure the traceability of fish products [8] and to distinguish farmed and non-farmed fish belonging to the same species [12].

In addition to landmarks, semilandmarks—not discrete anatomical loci—may also provide useful information, avoiding the use of some problematic landmarks. An example of a problematic landmark is the “focus” of the scale which is not visible in regenerated scales, which are very common in farmed fish [12].

Outline-based geometric morphometry (OL-based GM) is based on elliptic Fourier analysis used to compare shapes. Approaches to the study of scale shape using elliptic Fourier analysis have been applied for many years [15], and have been effectively used in discriminating fish populations [12,15,16,17], and in the description of the shape of scales from different body regions within the same species [18,19]. 

The aim of this study is to assess and compare two distinct geometric morphometry methods—specifically, the landmark-based and the outline-based approaches. Besides the main comparison, that is focused on distinguishing among species, we tried to broadly exploit our samples by attempting two minor aims: the comparison and discrimination among fish of the same species sampled from different areas (i.e., different aquaculture facilities), and of different body areas of a given species. This will be achieved through the analysis of a dataset comprising teleost elasmoid scales. The objective is to identify the most effective procedure that, with the implementation of computerized algorithms for analyzing large datasets, will minimize the bias associated with human operators. 

## 2. Materials and Methods

### 2.1. Sampling

The elasmoid scales analyzed in this study were collected from various specimens belonging to five different species of teleosts: the striped red mullet *Mullus surmuletus* (N = 3), the European seabass *Dicentrarchus labrax* (N = 2), the European pilchard *Sardina pilchardus* (N = 4), the gilthead seabream *Sparus aurata* (N = 18), and the zebrafish *Danio rerio* (N = 10). The specimens of *M. surmuletus*, *D. labrax*, and *S. pilchardus* were bought in a fish market in Genoa, Italy, and their origin was the Ligurian Sea. The 18 specimens of *S. aurata* considered in this study were obtained from different aquaculture facilities: aquaculture facility “Aqua De Mâ” (Lavagna–Genoa, Italy) (Ligurian sea), aquaculture facility Nutritech (Mantua, Italy), and another from an unspecified aquaculture facility in Turkey. The specimens from the first two aquaculture facilities had already been sacrificed for commercial purposes, and we collected some scales from the specimens destined for sale. The specimens from Turkey, on the other hand, were purchased at a supermarket. The specimens from Nutritech are indicated below as Nutritech1 and Nutritech 2 as they were from two different hatcheries before being reared at the Nutritech aquaculture facility. The specimens belonging to these four species were sampled and their total and standard lengths were measured (Table 1). 

The specimens of *D. rerio*, which were previously used in another project at the Department of Earth, Environmental, and Life Sciences (University of Genoa) and euthanized according to the ethics committee guidelines, were not measured but all of them were adults in the range of 2.5–3.5 cm. Scales from both sides of each specimen were collected from specific areas, as indicated in [20]. In total, 232 scales were analyzed. According to [21], scales were collected from the central areas of the body. In particular, scales were collected from the D area for all the species and from the C, E, and H areas only for *D. labrax* and *M. surmuletus* in order to compare the different body areas in these two species (Table 2). All scales were washed in water and accurately cleaned with a little brush before being analyzed.

Among the morphological features of a scale, the anterior margin exhibits considerable variability and plays a crucial role in analyzing scale shapes. Four out of the five selected species are common teleosts found in the Mediterranean Sea. These species were specifically chosen due to the distinct characteristics of their anterior scale margins, as outlined by [20]: *S. pilchardus*, *D. labrax*, *S. aurata*, and *M. surmuletus* possess scales with a smooth, striated, waved, and dentate anterior margin, respectively. This selection allowed for testing the methods on scales with these varying features. The fifth species is *D. rerio*, a freshwater teleost widely used as a model organism. Its scales show a smooth anterior margin. *D. rerio* was included in this study for two reasons: to increase the size and variability of the dataset and as an attempt to set a baseline for the geometric morphometry of its scales. Landmark-based geometric morphometry has already been used on *D. rerio* [22] but this is, to the best of our knowledge, the first time that the outline-based method considered in this study is used on its scales and we suggest it could be applied, in future works, on the study of scale morphogenesis and its disruption. 

### 2.2. Image Acquisition

Each sampled scale, preserved between two microscope slides stuck together by adhesive tape, was observed. Photographs were acquired through a stereomicroscope MZ APO (Leica, Wetzlar, Germany) equipped with a Moticam 10+ camera (Motic, Barcelona, Spain) using different magnifications according to the scale size. For *M. surmuletus*, the magnification was X6.3 (235 pixels/mm), for *D. rerio*, the magnification was X25 (932 pixels/mm), for all other species the magnification was X10 (371 pixels/mm). All scales were examined keeping the anterior portion leftwards and the dorsal portion upwards.

### 2.3. Geometric Morphometry Based on Landmarks and Semilandmarks (LM-Based GM)

Analyses were conducted considering five landmarks (positioned on five specific points common to all scales under consideration) and 80 semilandmarks placed along the scale’s edge. At the very beginning, a geometric morphometric analysis only using the five landmarks was performed. The PCA based on that analysis showed large overlapping among different species. Therefore, the method with five landmarks and 80 semilandmarks was preferred. The position of landmarks was chosen also considering the specific literature [8,9,10,13,21,22,23,24,25,26,27,28,29].

In *D. labrax* and in *S. aurata* the absence of a clear focus in regenerated scales has been described and, moreover, the occurrence of this kind of scale is indicated as very common in farmed specimens [12,30]. To make the method used here (LM-based GM method) widely appliable on farmed fish also, it was performed using only landmarks and semilandmarks positioned on the edge of the scales and without considering the position of the focus. 

In particular, as shown in Figure 1: dorsal and ventral anterior tips, respectively, corresponded to landmarks 1 and 2; dorsal and ventral posterior tips, respectively, corresponded to landmarks 5 and 3; and landmark 4 was located in the middle of the posterior margin of the scale. Forty semilandmarks were considered between landmarks 1 and 2 because this trait showed a higher morphological variability, while 10 semilandmarks were considered between the other couples of landmarks. 

Analyses based on landmarks and semilandmarks were conducted using ImageJ (version 1.53k) for image processing and R (version 4.1.2 “Bird Hippie”) for actual geometric morphometry. Procrustes analysis, Principal Components Analysis (PCA), and PERMANOVA cluster tests were carried out. PCA graphics were realized considering the first two variables of the multivariate set (the most representative) and using Microsoft PowerBI (Version July 2021). As well as geomorph (version 4.0.1) [31,32], the package vegan (version 2.6.4) was also used for cluster testing [33]. 

The centroid size for each different groups (i.e., from different aquaculture facilities) of *S. aurata* and for each species (considering only the scale from the region D) was calculated and then the shape-on-size regression was tested in order to highlight a possible allometry.

### 2.4. Outline-Based Geometric Morphometrics (OL-Based GM)

Photographs of scales were processed using the free software Inkscape (version 0.92.2 5c3e80d, 6 August 2017) [34] in order to obtain, for each scale, a black silhouette on a white background. The silhouettes were analyzed using the R package Momocs (version 1.4.1) [35]. The outcoming outlines were oriented and scaled in order to have the same orientation and centroid size (Figure 2). Elliptical Fourier transform (EFT), with 99% harmonic power, was performed on the whole dataset, on intraspecific subsets, and on pairwise subsets. PCA analysis, based on EFT, was performed for the dataset and all subsets. The R package vegan [33] was used for cluster tests. All the scripts used are available in Appendix A.

## 3. Results

### 3.1. Discrimination of S. aurata from Different Aquaculture Facilities

Scales from the D body area of 18 *S. aurata* specimens, from four different groups (Lavagna (Ligurian Sea), Nutritech 1, Nutritech 2 (Adriatic Sea) and Turkey) were analyzed (Table 1).

The Procrustes superimposition was performed to proceed with the LM-based GM and allowed to obtain the consensus coordinates of LMs and SLMs for this subset of scales (Figure 3a). The consensus coordinates are reported in Appendix A.

The stacking of centered and scaled outlines was performed before the OL-based GM and highlights a similar variability for all the four sides of the scales (Figure 3b).

The centroid size of the four groups was significantly different between Nutritech 1 and Lavagna, and between Nutritech 1 and Nutritech 2 (*p*-values 0.0058 and 0.0137 respectively). Nevertheless, the shape-on-size regression did not show a correlation (R^2^ = 0.008), suggesting that allometry can be overlooked, at least in this dataset.

The PCA was performed using both the Procrustes superimposition data and the EFT (Figure 4a and Figure 4b, respectively). The PC1 was mainly affected by the position of the LMs in the posterior margin and, in general, by the changes in the proportion of the dorso-ventral and the rostro-caudal axes of the scales; the PC2 was affected by the proportion of the anterior and the posterior margin length (Figure 5a,b).

According to the PERMANOVA, the scales of *S. aurata* from the different aquaculture facilities did not show significant clustering in the PCA from LM-based GM (adjusted *p*-values: see Table 3). However, the “Nutritech 1” and “Nutritech 2” groups were significantly clustered against the “Lavagna” group (both with an adjusted *p*-value < 0.01) in the OL-based GM, and, in particular the “Nutritech 2” and “Lavagna” pair shows an R^2^ of 0.50 (Table 3).

### 3.2. Discrimination of Different Body Areas in D. labrax and M. surmuletus

From the C, D, E, and H body areas, 59 scales from two *D. labrax* specimens and 53 scales from three *M. surmuletus* specimens were analyzed. The Procrustes superimposition was performed to proceed with the LM-based GM and allowed to obtain the consensus coordinates of landmarks and semilandmarks for *D. labrax* (Figure 6a) and *M. surmuletus* (Figure 6b). The consensus coordinates are reported in Appendix A. The stacking of centered and scaled outlines was performed before the OL-based GM for *D. labrax* scales (Figure 6c) and for *M. surmuletus* scales (Figure 6d).

The PCA was performed using both the Procrustes superimposition data and the EFT of *D. labrax* scales (Figure 7a and Figure 7b, respectively) and of *M. surmuletus* (Figure 8a and Figure 8b, respectively). In both *D. labrax* and *M. surmuletus*, the PC1 was affected by the LMs in the posterior and in the dorsal and ventral edges, by the changes in the proportion of the dorso-ventral and the rostro-caudal axes of the scales, and of the smoothening of the curve of the posterior margin. The PC2 was affected in *D. labrax* by the proportion of the anterior margin length and the rest of the shape, and in *M. surmuletus* by the smoothening of the general shape of the scales, mainly in the dorsal, ventral, and posterior edges (Figure 5c–f).

In *D. labrax*, the PERMANOVA performed on the PCA from LM-based GM showed significant differences among all the body areas (adjusted *p*-value < 0.01 for the couples D-C, D-E, D-H, and E-H, and adjusted *p*-value < 0.05 for the couples C-E and C-H). The highest R^2^ value (0.45) was between scales from the E and H regions. The PERMANOVA analysis performed on the PCA from OL-based GM showed significant differences among some of the body areas (adjusted *p*-value < 0.01 for the couples D-E, D-H, and E-H, and adjusted *p*-value < 0.05 for the couple C-H), with quite low R^2^ values (Table 4).

In *M. surmuletus*, the PERMANOVA performed on the PCA from both the GM methods, highlighted the same differences in term of significance. The couples D-C, D-E, and D-H were significantly different with an adjusted *p*-value < 0.01; the couples C-H and E-H were significantly different with an adjusted *p*-value < 0.05; the scales from the body areas C and E were not significantly different. The highest R^2^ values, from both methods, were from the C-H and E-H couples (Table 4).

### 3.3. Discrimination of the Species

The scales from the D body area of all the specimens were firstly analyzed pairwise for the 10 possible couples of species. For each couple of species, Procrustes superimposition of landmarks and semilandmarks was performed to proceed with the LM-based GM, while the stacking of centered and scaled outlines was performed before the OL-based GM (Figure 9). The PCA was performed using both the Procrustes superimposition data and the EFT (Figure 10).

The PERMANOVA performed on the PCA from both the GM methods highlighted significant differences between the scales of all the couples (all the *p*-values < 0.005). The R^2^ values were between 0.34 and 0.70 for each couple at least for one of the methods, except for the couple *D. rerio*-*S. pilchardus* that reached a R^2^ of 0.24 with the LM-based GM (Table 5).

The centroid size of the species was significantly different among *D. rerio* and all the other species (the scales of *D. rerio* were the smallest; all the *p*-values = 0.0000) and *M. surmuletus* and all the other species (the scales of *M. surmuletus* were the largest; all the *p*-values = 0.0000 but that between *M. surmuletus* and *S. pilchardus* = 0.0395). Nevertheless, the shape-on-size regression did not show a correlation (R^2^ = 0.117). The PCA was performed using both the Procrustes superimposition data and the EFT (Figure 11). 

The PERMANOVA performed on the PCA from both the GM methods and for both the datasets highlighted significant differences among the scales of all the species (all the adjusted *p*-values < 0.01) with R^2^ substantially similar to the pairwise comparison.

The PC1 was affected especially by the LMs in the posterior and in the dorsal and ventral edges, while the PC2 was affected by the LMs in the posterior and ventral edges (Figure 5g,h).

## 4. Discussion

The present study reports the results of two different methods of geometric morphometry, using the R packages geomorph [31,32] and Momocs [35] applied to teleost fish scales. This aims to improve the applicability of geometric morphometry in the forensic field, where it is necessary to identify a fish species quickly and easily. This approach could be considered preliminary, followed by molecular verifications, or it could be applied when the entire animal is not available for recognition based on taxonomic characteristics or, again, to enable recognition by individuals not particularly trained in fish taxonomy.

Five different fish species were selected for their commercial or scientific importance, and for the morphological features of their scales. Scales from four different body areas were sampled in two of the species, while specimens of one species (*S. aurata*) were from different aquaculture facilities. This experiment was mainly focused on distinguishing one fish species from another. Nevertheless, some other observations were carried out. 

The former experiment, conducted considering nominally different groups of *S. aurata*, as specimens came from different aquaculture facilities and/or hatcheries, did not give good results. Although the OL-based GM method appeared to be slightly more sensitive than the LM-based one, basically, the groups were not satisfactorily divided by the GM methods. In the literature, better results were obtained in the discrimination of real populations and phenotypes [12,14,36], which was not the case for our specimens.

Another secondary goal of this work was an attempt to discriminate the scales from different body areas of both *D. labrax* and *M. surmuletus*. The fact that the scales of each fish are not identical to one another and that they can assume different morphological characteristics along the body of the animal is well known [20]. Nevertheless, few studies approach the study of scale shape along the fish body by using at least one GM method [18,19]. In our study, both methods, LM-based and OL-based, gave positive results and some possible indications about the variation of scale shape along the fish body. In both species, the OL-based GM was not able to distinguish scales from the C and E body areas (see Figure 7 and Figure 8 for body areas). On the other hand, the scales from the C and E areas were significantly different in *D. labrax* but not in *M. surmuletus* when analyzed using the LM-based GM, although with a quite low R^2^ (0.23). For almost all the couples, the R^2^ values were higher using the LM-based GM (Table 4). This suggests that, for this specific task, the LM-based GM turned out to be more sensitive than the OL-based GM method. Also in this case, further analyses are necessary to better understand how to exploit the GM methods in the study of scale shape variation along the fish body, possibly correlating this variation to functional differences.

The main goal of this work was to compare the two GM methods in discriminating among some teleost fish species. Overall, the results show that both the LM-based and OL-based GM methods used here are reliable in distinguishing among the analyzed teleost species. 

Although this is not the first work stating the possibility of using GM on fish scales to discriminate teleost species in various context, it presents some important observations that could contribute to a better exploitation of the GM methods for the geometric scale morphometry of teleosts. Previous studies that applied LM-based GM to fish scale, considered from 6 to 15 landmarks and considered no semilandmarks. 

The most used landmark configuration takes into account seven landmarks: the four points at the limits among the anterior, dorsal, posterior, and ventral margins of the scale, the two points in the middle of the anterior and posterior margins, and the focus [8,9,12,13,21,22,24,25,26,27,28,36,37,38]. Few studies considered six landmarks, i.e., the same indicated above with the exception of the focus [10,23]. Grady and colleagues [39], working on species with a homogeneous kind of scale, all belonging to darters fish, used a unique seven-landmark configuration. Compared to the previous described seven-landmark configuration, they did not consider the middle point on the anterior margin but included a point at the base of the central ctenus, always recognizable in the scales analyzed belonging to that specific group of fish. In a similar way, working on quite a homogeneous kind of scale belonging to cichlids, Albertson and colleagues [29] detected eight landmarks, overlooking the focus as well as the limits of the posterior margin of the scale, and adding two points at the limits of the anterior radii. Working on a single species, the Arctic charr *Salvelinus alpinus*, Garduño-Paz and colleagues [14] reached the number of nine landmarks, by adding to the most common seven-landmark configuration two more landmarks in two specific points of the dorsal and ventral margins. Recently, a different and interesting approach to landmark positioning used the method of drawing 14 lines in a sunburst from the scale focus and then positioning a landmark where each line intercepts the margin of the scale. Those 14 landmarks plus the focus itself made the final 15-landmark configuration [40].

Although an approach with only five landmarks would not allow distinguishing among species, the five-landmarks-plus-80-semilandmarks method proposed in the present study seems to give good results in species discrimination and we believe it could make the procedure somehow more generally applicable and easier.

As known, in LM-based GM, landmarks should be chosen to be homologous among different specimens, while semilandmarks are not individually homologous even though they are points along homologous curves [41]. In this study, the number of semilandmarks for each of the four margins of the scale was pre-established, while their position was calculated by the software to make them equidistant to one another on each margin. In general, the positioning of the semilandmarks, as far as the curves (in this case the margins of the scales) are homologous, does not raise any homology-related issue. Therefore, introducing 80 semilandmarks does not raise major problems. On the other hand, reducing the number of landmarks might solve some of them.

The five landmarks proposed in this study are also present in the common seven-landmark configuration: they are the four points at the limits among the anterior, dorsal, posterior, and ventral margins of the scale, and the point in the middle of the posterior margin. Thus, two landmarks were discarded from the common seven-landmark configuration to obtain the five-landmark configuration.

One of the two discarded landmarks is the middle point along the anterior margin of the scale. The basic idea of the LM-based GM is that landmarks should be homologous points, undoubtably recognizable on the studied structures [41], and from this point of view, the anterior margin of a fish scale could sometimes be puzzling. In fact, the anterior margin of a fish scale can be smooth or can often present some oscillations, which have different names (e.g., waved, scalloped, dentate) and which are quite large compared to the size of the scale. Thus, they are not negligible when positioning landmarks (See Figure 1). On the other hand, the posterior margin can present spines called cteni but, in most of the species, cteni are quite small compared to the scale size and do not affect the general shape of the scale itself [20]. So, by discarding the middle point of the anterior margin, we remove troubles for the operator and the possibility to introduce a mistake due to the irregular oscillation of the scale margin. The semilandmarks we use along all the margins (they are more numerous along the anterior margin because of its often wavy trend) do not present problems, as they are equidistantly positioned by the software along the margin and, if numerous enough, they describe the margin shape.

The second landmark that is often present in the literature and that we discarded is the focus of the scale. The focus is a proper homologous landmark but, as stated before, it is not always clearly visible. Particularly in regenerated scales, which are a good percentage of the scales in fish from aquaculture [30], the focus is not visible at all. By removing the scale focus from our analysis, we obtained a method that could also be applied to regenerated scales. This could be relevant, for example, in commercial and forensic contexts.

In the literature, the OL-based GM has been generally less used than the LM-based GM while studying fish scales [12,16,17,18,19]; in particular, the Momocs package [35] has never been used for this purpose. We demonstrated here that, at least for our samples, the LM-based GM method (with five landmarks and 80 semilandmarks) performed using the geomorph package in R [31,32] and the OL-based GM method performed using the Momocs package in R [35] work equally well in discriminating species. The R^2^ values are generally higher using the LM-based GM, but that is not true for all the couples of species. Nevertheless, the OL-based GM seems to be more sensitive for some couples of species. We consider that they are good values overall in light of a low number of scales considered here (Table 2). 

Our results are in accordance with at least one other study that compared the LM-based and OL-based methods as well (although not using the same packages and setting) on scales from different *S. aurata* populations, obtaining better results in discriminating populations by using OLs than by using LMs [12]. We should consider that we compared few *S. aurata* specimens. Nevertheless, slightly better results were obtained using OLs than LMs (see Table 3).

In light of these results, and in the perspective of automating a major part of the algorithm necessary to detect the correct fish species by analyzing scales, we suggest that working with the OL-based GM would be convenient. In some contexts, like in the forensic one, the possibility to automate the analyses would also make them feasible for non-specialized workers. The positioning of landmarks requires a better understanding of the scale morphology, both by a human operator or by software. On the other hand, tracing the OL of a flat object is an easier task and, again, is independent of the scale focus. The recently published method, discussed above, that uses landmarks obtained from the positioning of equally spaced lines, goes in the direction of making the understanding of the structure of the scale unnecessary. Nevertheless, that method is not focus independent [40].

If the proposed methods appear promising for well-preserved scales, some issues may arise with lower-quality scales (e.g., those damaged by fishing nets or partially digested by predators). The problem of missing data is crucial in morphometric techniques. In some cases, reducing the number of landmarks is used to address this issue, but it can result in the loss of shape information. However, in our case, due to the limited number of landmarks, this approach is not feasible. Another common solution, mirroring objects along a midplane to discard damaged parts, is ineffective for objects with asymmetry, such as scales. Several methods are available in the literature to estimate missing landmarks [42], and some of them, such as thin-plate spline interpolation, also work for missing curves or semilandmarks [43]. While a detailed discussion on the problem of missing data in geometric morphometry is beyond the scope of the present study, it will need to be addressed in the future, especially considering that methods for fish species recognition based on scale shapes could be particularly useful in contexts where damaged scales are common.

## 5. Conclusions

In conclusion, this study confirms some previous results on the possibility of using the GM of fish scales for teleost species discrimination. It validates the Momocs package [35] for the outline analysis of fish scales, and points out some new considerations about the use of GM methods on fish scales for species discrimination purposes.

## Figures and Tables

**Figure 1 animals-14-01090-f001:**
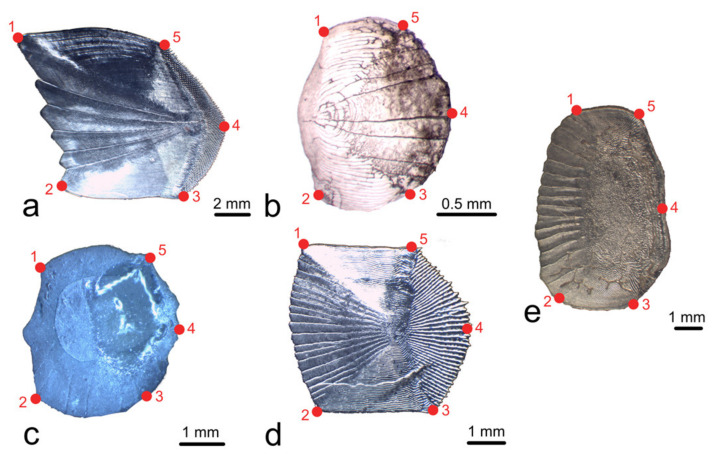
The red numbered dots show the position of the five landmarks for all the scales belonging to the five species considered: *Mullus surmuletus* Linnaeus, 1758 (**a**), *Danio rerio* (Hamilton, 1822) (**b**), *Sardina pilchardus* (Walbaum, 1792) (**c**), *Dicentrarchus labrax* (Linnaeus, 1758) (**d**), and *Sparus aurata* Linnaeus, 1758 (**e**).

**Figure 2 animals-14-01090-f002:**
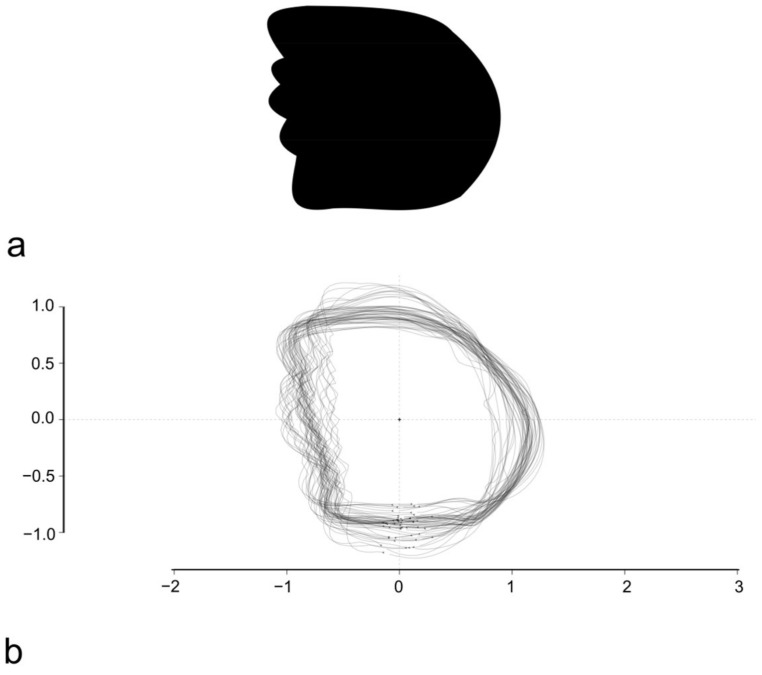
Example of a silhouette of a single *M. surmuletus* scale (**a**) and the result of the overlapping of all *M. surmuletus* scales scaled (**b**).

**Figure 3 animals-14-01090-f003:**
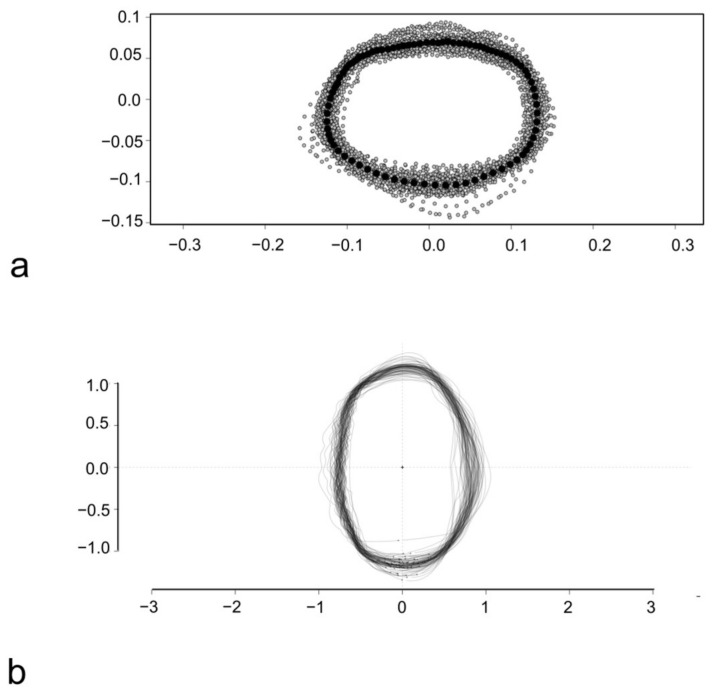
Procrustes superimposition of *S. aurata* scales (**a**) and overlapping of centered and scaled outlines of *S. aurata* scales (**b**). In (**a**) the anterior margin of the scale is above, and the dorsal margin is rightward. In (**b**) the anterior margin is leftward, and the dorsal margin is above.

**Figure 4 animals-14-01090-f004:**
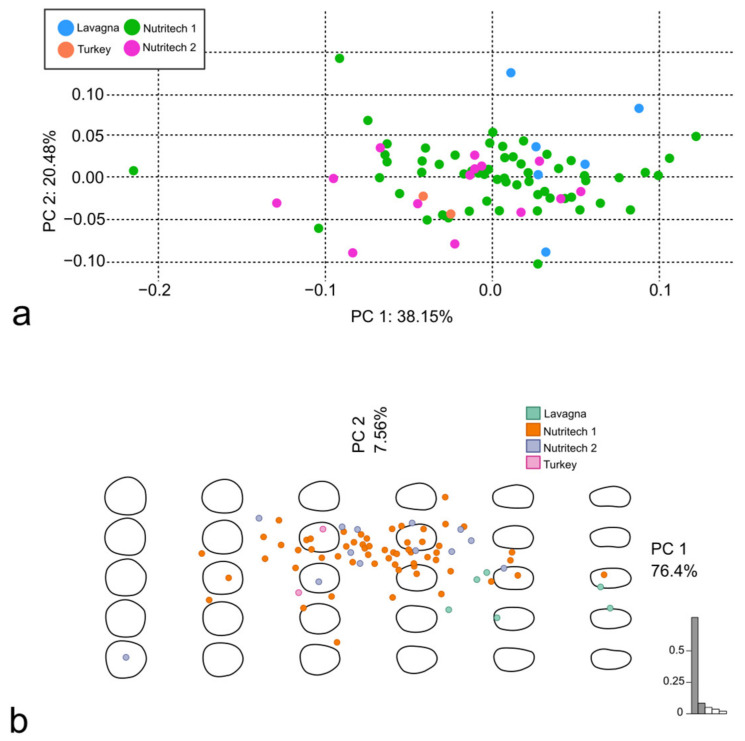
PCA performed using Procrustes superimposition data (**a**) and PCA performed using EFT (**b**), both on scales belonging to *S. aurata*.

**Figure 5 animals-14-01090-f005:**
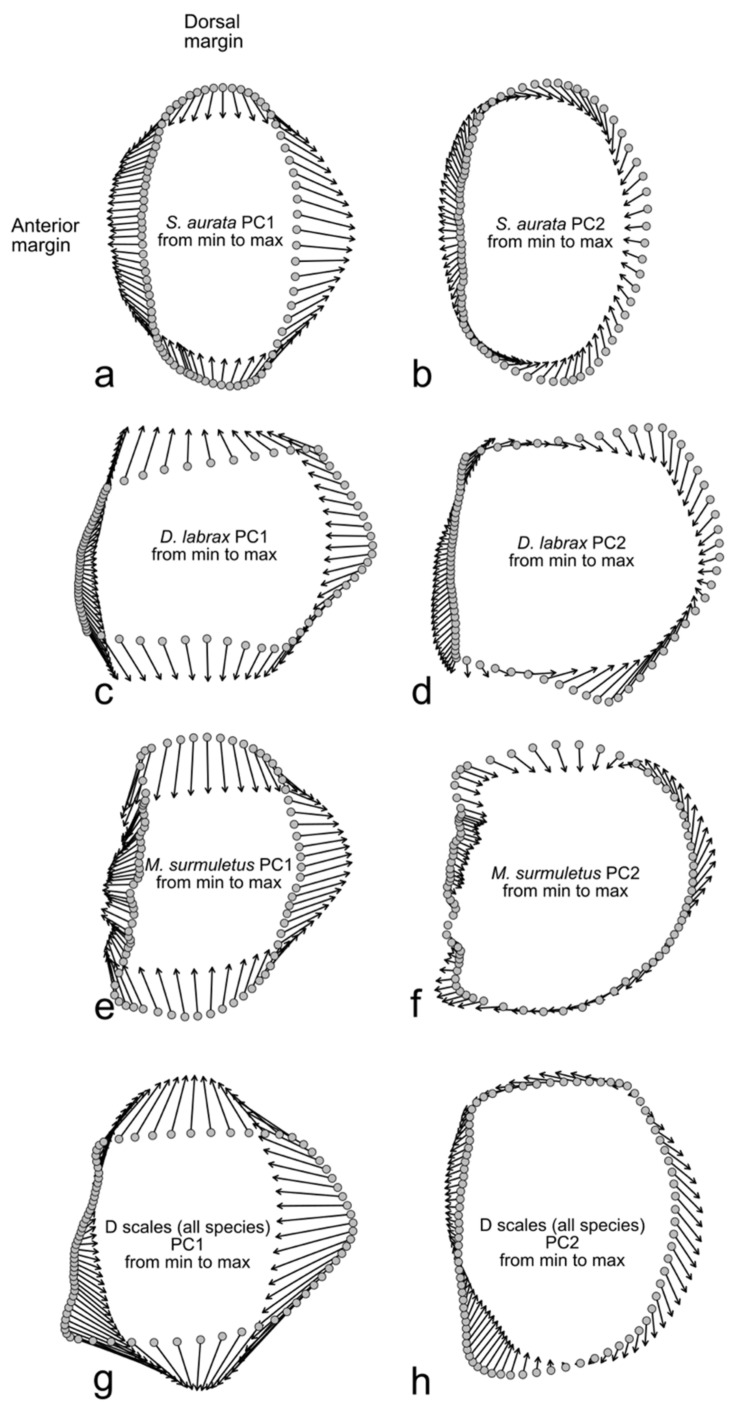
After the PCA performed using Procrustes superimposition data, vectors indicate the shape differences along the PC1 axis (**a**,**c**,**e**,**g**) and PC2 axis (**b**,**d**,**f**,**h**) for the PCA on *S. aurata* scales (**a**,**b**), on *D. labrax* scales (**c**,**d**), on *M. surmuletus* scales (**e**,**f**), and on the scales of all the species (**g**,**h**).

**Figure 6 animals-14-01090-f006:**
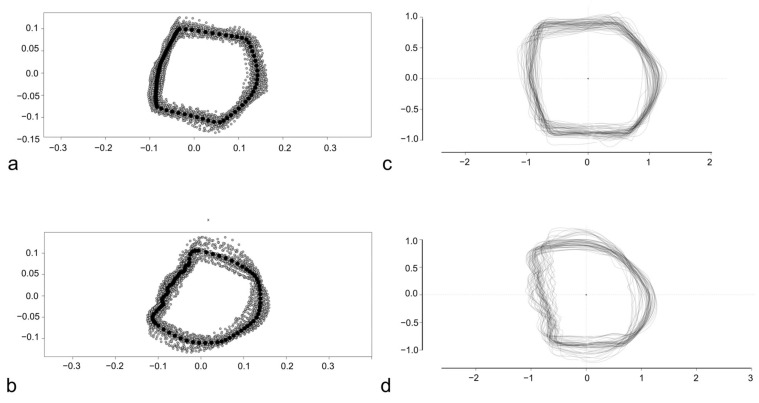
Procrustes superimposition of *D. labrax* (**a**) and *M. surmuletus* scales (**b**) and overlapping of centered and scaled outlines of *D. labrax* (**c**) and *M. surmuletus* (**d**) scales.

**Figure 7 animals-14-01090-f007:**
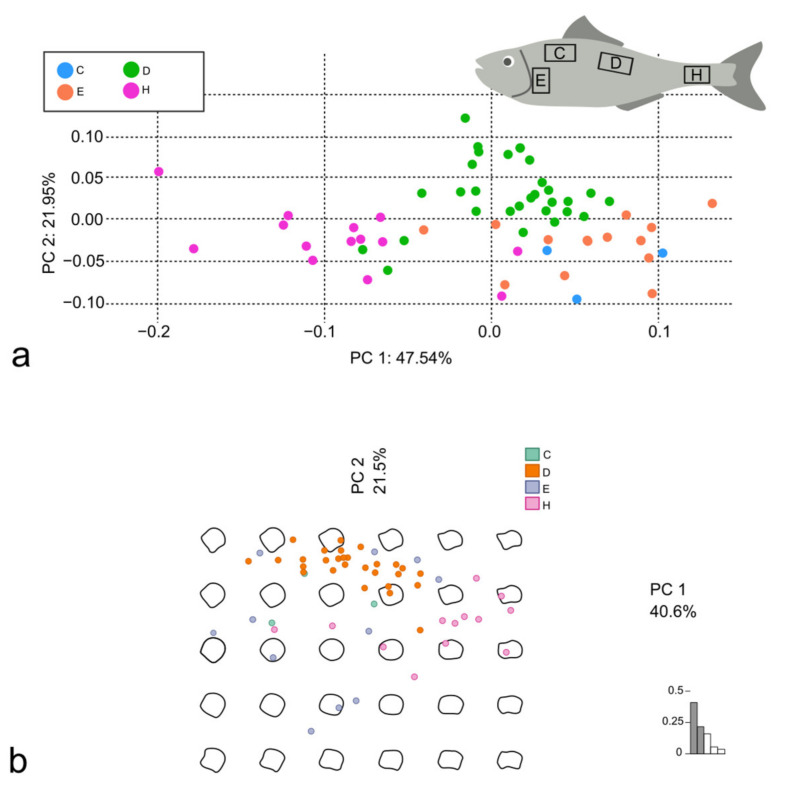
PCA performed using Procrustes superimposition data (**a**) and PCA performed using EFT (**b**), both on scales belonging to *D. labrax*.

**Figure 8 animals-14-01090-f008:**
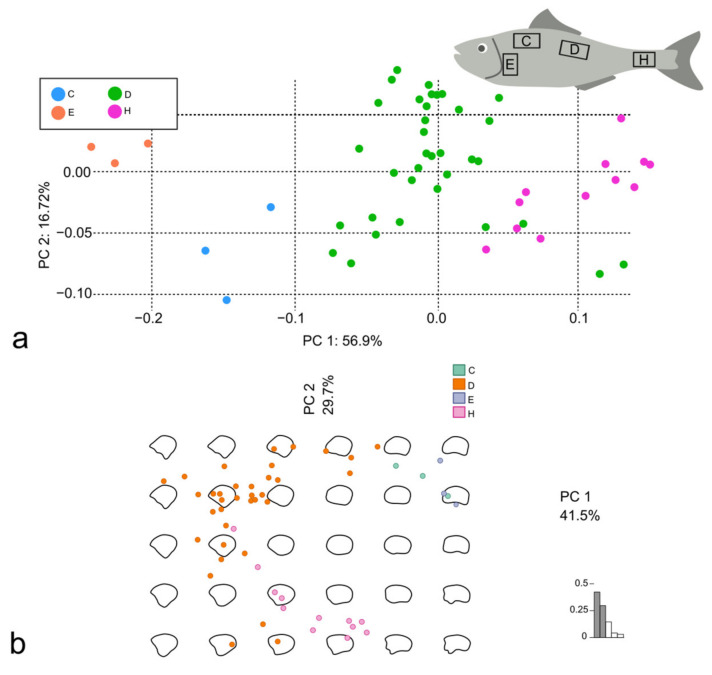
PCA performed using Procrustes superimposition data (**a**) and PCA performed using EFT (**b**), both on scales belonging to *M. surmuletus*.

**Figure 9 animals-14-01090-f009:**
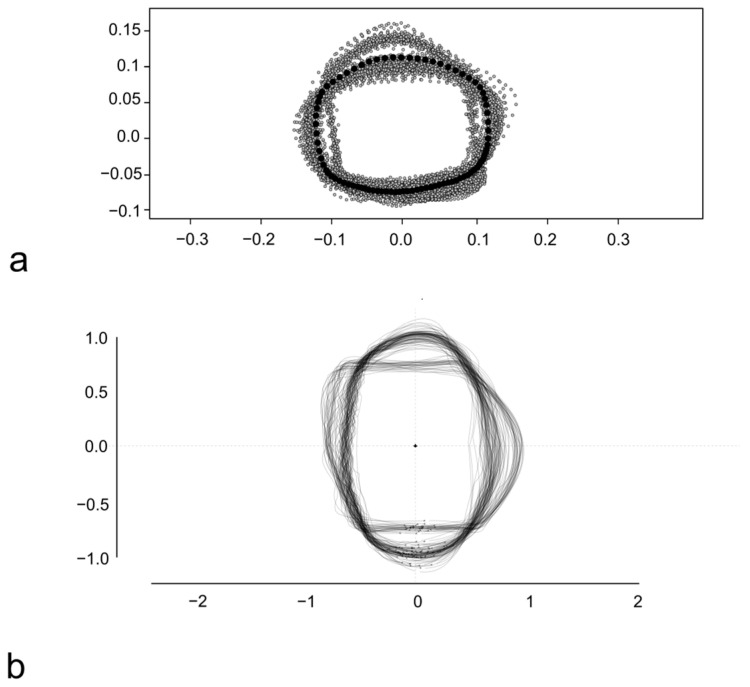
Example of the comparison between LM-based GM and OL-based GM for a couple of species. Procrustes superimposition (**a**) and overlapping of centered and scaled outlines (**b**) of *D. labrax* and *S. aurata* scales.

**Figure 10 animals-14-01090-f010:**
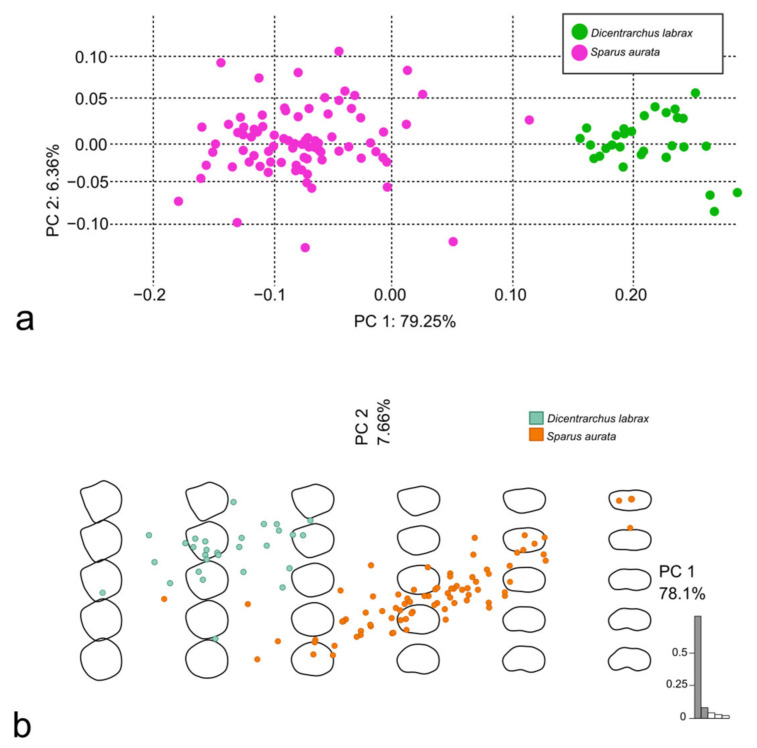
Comparison between scales belonging to *D. labrax* and *S. aurata*. PCA performed using Procrustes superimposition data (**a**) and PCA performed using EFT (**b**).

**Figure 11 animals-14-01090-f011:**
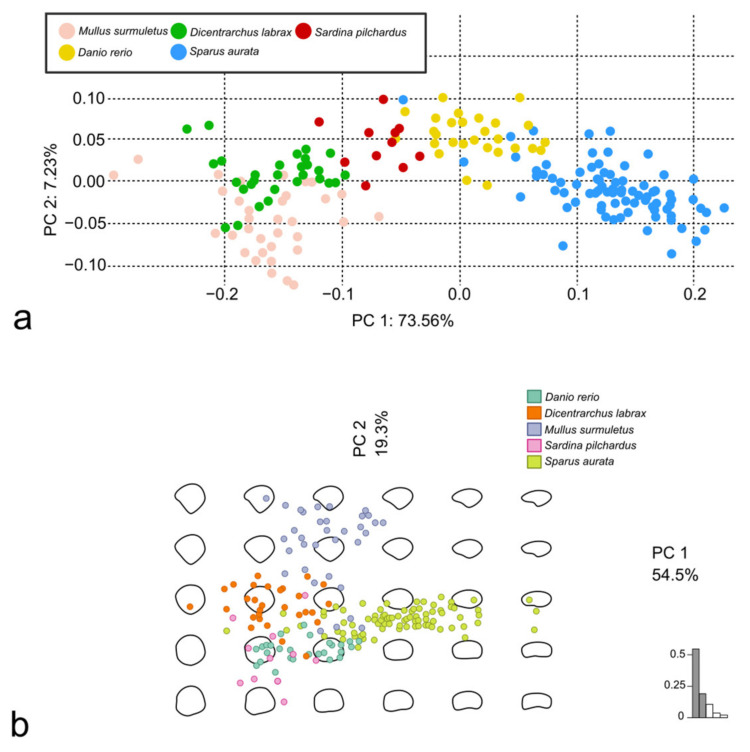
PCA performed using Procrustes superimposition data (**a**) and PCA performed using EFT (**b**), both on scales belonging to *M. surmuletus*, *D. labrax*, *S. pilchardus*, *S. aurata,* and *D. rerio*.

**Table 1 animals-14-01090-t001:** Total and standard lengths measured for each directed sampled specimen.

Species	Specimens	Total Length (cm)	Standard Length (cm)	Source
*Mullus surmuletus* (Linnaeus, 1758)	Mulsur1	21.2	19.2	Professional fisheries
	Mulsur2	25.0	21.0	Professional fisheries
	Mulsur3	22.4	18.7	Professional fisheries
*Dicentrarchus labrax* (Linnaeus, 1758)	Diclab1	34.6	29.0	Professional fisheries
	Diclab2	32.9	28.8	Professional fisheries
*Sardina pilchardus* (Walbaum, 1792)	Sarpil1	12.1	10.4	Professional fisheries
	Sarpil2	11.9	10.3	Professional fisheries
	Sarpil3	11.5	9.9	Professional fisheries
	Sarpil4	10.6	9.1	Professional fisheries
*Sparus aurata* (Linnaeus, 1758)	Spaaur21.1	26.5	23.9	Lavagna
	Spaaur21.2	27.4	23.5	Turkey
	Spaaur21.3	18.6	15.6	Nutritech 1
	Spaaur21.4	17.5	14.5	Nutritech 1
	Spaaur21.5	19.1	16.2	Nutritech 1
	Spaaur21.6	17.8	15.4	Nutritech 1
	Spaaur21.7	23.8	20.7	Nutritech 2
	Spaaur21.8	23.1	19.6	Nutritech 2
	Spaaur21.9	22.4	18.6	Nutritech 2
	Spaaur21.10	22.5	18.7	Nutritech 2
	Spaaur21.11	23.8	19.9	Nutritech 1
	Spaaur21.12	25.2	22.1	Nutritech 1
	Spaaur21.13	22.7	20.2	Nutritech 1
	Spaaur21.14	27.5	24.2	Nutritech 1
	Spaaur21.15	28.5	24.7	Nutritech 1
	Spaaur21.16	27.6	24.3	Nutritech 1
	Spaaur21.17	30.2	26.7	Nutritech 1
	Spaaur21.18	34.2	29.1	Nutritech 1
*Danio rerio* (Hamilton, 1822)	10 specimens	2.5–3.5	Not recorded	University of Genoa

**Table 2 animals-14-01090-t002:** Number of scales and sampling areas for each species.

Species	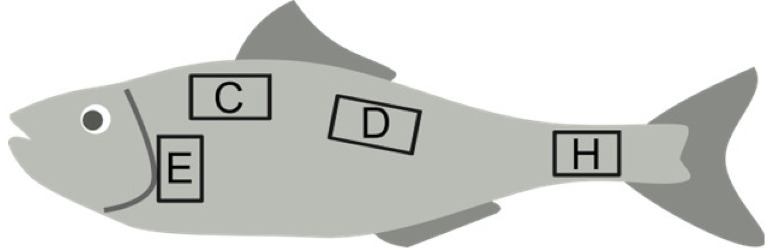	Sampling Area			
		C	D	E	H
*Mullus surmuletus* (Linnaeus, 1758)		3	35	3	12
*Dicentrarchus labrax* (Linnaeus, 1758)		3	29	13	14
*Sardina pilchardus* (Walbaum, 1792)			11		
*Sparus aurata* (Linnaeus, 1758)			81		
*Danio rerio* (Hamilton, 1822)			28		

**Table 3 animals-14-01090-t003:** R^2^ and adjusted *p*-values from the PERMANOVA in pairwise comparisons of different aquaculture facilities of *S. aurata*. R^2^ is bold if ≥0.3; the gray background highlights the best result between the landmark-based geometric morphometry (LM-based GM) and the outline-based geometric morphometry (OL-based GM), when the differences are significant. * = *p*-Value < 0.05, ** = *p*-Value < 0.01, L = Lavagna, N1 = Nutritech1, N2 = Nutritech2, T = Turkey.

Pairs	LM-Based GM		OL-Based GM	
	R^2^	Adj. *p*-Value	R^2^	Adj. *p*-Value
T vs. N1	0.03	0.534	0.04	0.642
T vs. N2	0.10	0.852	0.08	1.000
T vs. L	0.25	0.816	**0.62**	0.234
N1 vs. N2	0.02	0.726	0.05	0.234
N1 vs. L	0.05	0.120	0.17	0.006 **
N2 vs. L	0.20	0.012 *	**0.50**	0.006 **

**Table 4 animals-14-01090-t004:** R^2^ and *p*-values from the PERMANOVA in comparisons of scales from different body areas of *D. labrax* and *M. surmuletus*. R^2^ is bold when ≥0.3; the gray background highlights the best result between the landmark-based geometric morphometry (LM-based GM) and the outline-based geometric morphometry (OL-based GM), when the result is significant. * = *p*-Value < 0.05, ** = *p*-Value < 0.01.

Pairs		LM-Based GM		OL-Based GM			LM-Based GM		OL-Based GM	
		R^2^	Adj. *p*-Value	R^2^	Adj. *p*-Value		R^2^	Adj. *p*-Value	R^2^	Adj. *p*-Value
D vs. C	*D. labrax*	0.23	0.006 **	0.08	0.222	*M. surmuletus*	0.21	0.006 **	0.23	0.006 **
D vs. E		0.26	0.006 **	0.16	0.006 **		**0.33**	0.006 **	0.28	0.006 **
D vs. H		**0.34**	0.006 **	0.28	0.006 **		0.25	0.006 **	0.26	0.006 **
C vs. E		0.23	0.018 *	0.05	1.000		**0.54**	0.600	**0.40**	0.600
C vs. H		**0.38**	0.030 *	0.24	0.030 *		**0.69**	0.012 *	**0.51**	0.012 *
E vs. H		**0.45**	0.006 **	0.22	0.006 **		**0.79**	0.018 *	**0.53**	0.024 *

**Table 5 animals-14-01090-t005:** R^2^ and *p*-values (pairwise) or adjusted *p*-values (more than two species compared) from the PERMANOVA in pairwise comparisons of all the species, in comparison among all the species at the same time, and in comparison among all the marine species at the same time. R^2^ is bold when ≥0.3; the gray background highlights the best result between the landmark-based geometric morphometry (LM-based GM) and the outline-based geometric morphometry (OL-based GM). The species are indicated with the first letters of their names, e.g., Dl = *Dicentrarchus labrax* and so forth. * = *p*-Value < 0.05, ** = *p*-Value < 0.01, *** = *p*-Value < 0.005.

Pairs		LM-Based GM		OL-Based GM			LM-Based GM		OL-Based GM			LM-Based GM		OL-Based GM	
		R^2^	*p*-Value	R^2^	*p*-Value		R^2^	Adj. *p*-Value	R^2^	Adj. *p*-Value		R^2^	Adj. *p*-Value	R^2^	Adj. *p*-Value
Dl/Sa	Pairwise	**0.70**	0.001 ***	**0.48**	0.001 ***	Altogether	**0.70**	0.01 *	**0.48**	0.01 *	Altogether without *D. rerio*	**0.70**	0.006 **	**0.48**	0.006 **
Dl/Sp		**0.36**	0.001 ***	**0.30**	0.001 ***		**0.35**	0.01 *	**0.30**	0.01 *		**0.35**	0.006 **	**0.30**	0.006 **
Dl/Ms		0.20	0.001 ***	**0.35**	0.001 ***		0.19	0.01 *	**0.36**	0.01 *		0.19	0.006 **	**0.36**	0.006**
Dl/Dr		**0.62**	0.001 ***	**0.37**	0.001 ***		**0.61**	0.01 *	**0.37**	0.01 *					
Sa/Sp		**0.38**	0.001 ***	**0.32**	0.001 ***		**0.39**	0.01 *	**0.32**	0.01 *		**0.39**	0.006 **	**0.32**	0.006 **
Sa/Dr		**0.34**	0.001 ***	**0.36**	0.001 ***		**0.34**	0.01 *	**0.36**	0.01 *					
Sa/Ms		**0.70**	0.001 ***	**0.34**	0.001 ***		**0.71**	0.01 *	**0.34**	0.01 *		**0.70**	0.006 **	**0.34**	0.006 **
Sp/Ms		0.29	0.001 ***	**0.34**	0.001 ***		**0.31**	0.01 *	**0.34**	0.01 *		**0.31**	0.006 **	**0.34**	0.006 **
Dr/Ms		**0.57**	0.001 ***	**0.41**	0.001 ***		**0.59**	0.01 *	**0.41**	0.01 *					
Dr/Sp		0.24	0.001 ***	0.12	0.001 ***		0.26	0.01 *	0.12	0.01 *					

## Data Availability

Data available in a publicly accessible repository: https://doi.org/10.6084/m9.figshare.25393621.v1 (accessed on 12 March 2024).

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
