# Peer review of "New Insights into Geometric Morphometry Applied to Fish Scales for Species Identification"

_animals, 2024, doi:10.3390/ani14071090_

Round 1
Reviewer 1 Report
Comments and Suggestions for Authors
Dear Authors,
please find in the enclosed file my corrections to the manuscript. These especially regard the use of the correct zoological nomenclature rules. Tables can be improved. Please be aware that you are not considering populations but specimens from different areas. To describe a population high number of specimens (200 at least) are required. Please consider to highlight the fact that in your manuscript the comparison between populations is not possible with the data obtained in your study, but it is possible to make comparison between methods used.
All the best regards
The Reviewer

Reviewer 2 Report
Comments and Suggestions for Authors
This manuscript finely describes and compares the application of geometric and landmark-based morphometry with interesting outcomes on the identification of five very different fish species. I appreciated the average innovation of the topic; the results are well described and discussed. Nevertheless, I have a couple of points I wish to be considered by the authors and, if possible, briefly discussed.
1) Among the ultimate scopes of the methodology developed, the smoothing, via the automatisation, of the process seems to be promising on well-preserved scales. What about lower quality scales (e.g., those damaged from fishery nets or partially digested by a predator)?
2) This method appears very useful and it will be even better when applied on a larger dataset, but did you consider to test it on similar species or congenerics (e.g., Mullus surmuletus vs M. barbatus; Lophius budegassa vs L. piscatorius. Species identification of genus Lophius is usually performed with the section of the animal on-board - unless the fisherman is very skilled - and this may result in a lower quality of the commercial product). I believe this would be interesting to consider.
Lastly, here some minor comments.
Line 48: “aquatic foods” sounds weird. Better “seafood”? Please consider this suggestion and check throughout the manuscript.
From line 200 (Figure 3) to the end of manuscript (captions included): species name should be in italics; please, carefully check throughout the manuscript and be consistent.
The English form is satisfying.
I would thank the authors for their interesting work and I recommend minor revisions.
